# Representation Learning with Large Language Models for Recommendation

## ABSTRACT

Recommender systems have seen significant advancements with the influence of deep learning and graph neural networks, particularly in capturing complex user-item relationships. However, these graph-based recommenders heavily depend on ID-based data, potentially disregarding valuable textual information associated with users and items, resulting in less informative learned representations. Moreover, the utilization of implicit feedback data introduces potential noise and bias, posing challenges for the effectiveness of user preference learning. While the integration of large language models (LLMs) into traditional ID-based recommenders has gained attention, challenges such as scalability issues, limitations in text-only reliance, and prompt input constraints need to be addressed for effective implementation in practical recommender systems. To address these challenges, we propose a model-agnostic framework RLMRec that aims to enhance existing recommenders with LLM-empowered representation learning. It proposes a recommendation paradigm that integrates representation learning with LLMs to capture intricate semantic aspects of user behaviors and preferences. RLMRec incorporates auxiliary textual signals, develops a user/item profiling paradigm empowered by LLMs, and aligns the semantic space of LLMs with the representation space of collaborative relational signals through a cross-view alignment framework. This work further establish a theoretical foundation demonstrating that incorporating textual signals through mutual information maximization enhances the quality of representations. In our evaluation, we integrate RLMRec with state-of-the-art recommender models, while also analyzing its efficiency and robustness to noise data.

**ACM Reference Format:**
Anonymous Author(s). 2023. Representation Learning with Large Language Models for Recommendation. In *Proceedings of ACM Conference*. ACM, XXX, XXX, 12 pages. https://doi.org/10.1145/nnnnnnn.

## 1 INTRODUCTION

Recommender systems have evolved to provide personalized item recommendations based on user interactions, with deep learning and graph neural networks playing a significant role [4, 39]. Graph-based recommenders like NGCF [35] and LightGCN [11] have demonstrated impressive capabilities in capturing complex user-item relationships, making them state-of-the-art approaches.

However, it is important to acknowledge that recent graph-based recommenders heavily rely on ID-corresponding information for learning. In this line, the training data consists exclusively of mapped user/item indices, and their interactions are represented in an interaction matrix using binary values (1 indicating an interaction and 0 indicating no interaction). While this data arrangement has demonstrated effectiveness, one significant limitation is that

it primarily relies on ID-based information, thereby potentially overlooking other valuable data, such as rich textual information associated with users and items. The absence of this additional information can lead to reduced informativeness in the learned representations. Furthermore, it is worth noting that a substantial portion of the data in these graph-based recommenders consists of implicit feedback [27, 33], which can introduce noise from false negatives or bias (*e.g.*, misclicks [34] or popularity bias [5]). Consequently, the learned representations of these GNN-based models heavily rely on the inherent quality of the data. This heavy reliance on the data quality poses a potential challenge as it can lead to detrimental representations that hinder the effectiveness of recommendation systems, especially when the data contains noise.

In recent times, there have been several endeavors to leverage diverse data modalities in order to enhance traditional ID-based recommenders [9, 18, 45]. Particularly interesting is the emergence of large language models (LLMs) like GPT-4 [25] and LLaMA [32], which have demonstrated impressive capabilities in neural language understanding tasks. This development has sparked significant interest among researchers, who are actively exploring how LLMs, with their proficiency in handling textual content, can extend the capabilities of recommendation systems beyond the original data [7, 19, 22]. A primary focus of current research in this field revolves around aligning recommendation approaches with the characteristics of language models through prompt design. Methods like InstructRec [46] structure the recommendation task in an instruction-question-answering format, enabling LLMs to simultaneously address the recommendation objective and respond to intricately designed questions [1, 9]. However, despite displaying some recommendation capabilities, these methods still fall behind existing recommenders in terms of efficiency and precision. This can be attributed to inherent shortcomings associated with this approach, including the following key aspects:

i) **Scalability issues in practical recommenders**. The utilization of large language models (LLMs) in personalized user behavior modeling, comes with inherent computational demands. As the scale of user behavior data increases, the computational requirements and associated inference time costs also tend to rise significantly. For instance, in the case of TALLRec [1], where recommendations are generated based on an instruction-question-answering format, the response time for LLaMA2-13B to provide recommendations to individual users stands at approximately 3.6 seconds, based on an input size of around 800 tokens (equivalent to approximately 5 users). However, this poses significant challenges when attempting to scale up the LLM-based approach for practical recommender systems with a substantial user base and extensive item catalog.

ii) **Limitations stemming from text-only reliance**. LLMs have the potential to generate text answers that may include recommendations for non-existent items due to hallucination issues [21]. This poses a challenge in ensuring the accuracy and reliability of the

*ACM Conference, XXX, XXX*
2023. ACM ISBN 978-x-xxxx-xxxx-x/YY/MM...$15.00
https://doi.org/10.1145/nnnnnnn.

                                                                        

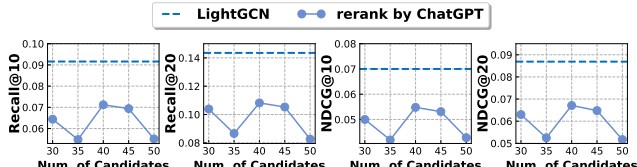

Figure 1: LLM's performance for recommendation reranking when dealing with different sizes of candidate items.

generated recommendations. Additionally, the limited capacity of prompt inputs, constrained by the maximum number of tokens (e.g., 2048 tokens for LLaMA), hinders the effective modeling of comprehensive collaborative signals with global user dependencies.

To validate the aforementioned limitations, we conduct an evaluation to assess the effectiveness of directly using LLM in enhancing the re-ranking task [12, 31] for recommendation on the Amazon dataset. Specifically, we utilize LightGCN [11] as the underlying backbone recommender model, which generate a ranking list of 50 candidate items preferred by each user based on learned user-item interaction probabilities. To further refine the recommendations, we integrate the textual information of each item using our custom prompts (for prompt format details, please refer to Appendix A A.3). These prompts are then processed by ChatGPT platform (*i.e.*, gpt-3.5-turbo). The objective of this evaluation is to identify the Top-10 and Top-20 most relevant items from the ranking list generated by LightGCN for each user through the re-ranking task.

It is evident from the results in Figure 1 that the recommendations refined by the ChatGPT perform worse than the original results provided by LightGCN. This indicates that there are limitations when blindly using LLMs to improve the re-ranking process in recommendation systems that heavily rely on textual information. These limitations can be attributed to three key factors: i) The hallucination issue of LLMs, where they may suggest recommended items that are not included in the candidate set; ii) The lack of a comprehensive global text-based collaborative relationship input due to the token limit of LLMs; iii) Additionally, it is worth noting that the reranking process using LLM takes several hours to complete, which poses a challenge when dealing with large-scale data in real-world recommendation scenarios. Due to page limit, we delve deeper into this experiment and present real cases to illustrate the hallucination phenomenon for the reranking in the Appendix.

**Contributions**. In light of the aforementioned limitations, our work aims to leverage the power of LLMs to seamlessly enhance existing recommender systems. To accomplish this, we propose a model-agnostic framework called RLMRec (Representation Learning with Large Language Models for Recommendation). The core idea of RLMRec is to utilize representation learning as a bridge between ID-based recommenders and LLMs. Our new recommendation paradigm aims to preserve the accuracy and efficiency of existing recommenders while harnessing the powerful text comprehension capabilities of LLMs to understand the intricate semantic aspects of user behaviors and preferences. To begin, we lay the theoretical groundwork by modeling the benefits of incorporating auxiliary textual signals for representation learning. This involves transforming the textual signals into meaningful representations and establishing a theoretical foundation for maximizing mutual information within general recommendation models. Furthermore,

we develop a user/item profiling paradigm empowered by LLMs. This paradigm enables us to incorporate the comprehensive semantic understanding derived from LLMs into the encoded representations of users and items. By leveraging the global knowledge space of LLMs, we enhance the representation expressiveness.

Furthermore, we propose to align the semantic space of LLMs and the representation space of collaborative relational signals through a cross-view alignment framework. This alignment is achieved through a cross-view mutual information maximization scheme, which allows us to find a common semantic subspace where the textual and collaborative relational embeddings are well aligned from the contrastive and generative modeling, respectively. This typically involves jointly optimizing the recommendation model parameters and the cross-modal alignment paradigm. In a nutshell, our main contributions can be summarized as follows:

- This work aims to explore the potential of enhancing the recommendation performance of existing recommender systems, by leveraging LLMs and aligning their semantic space with collaborative relation modeling for better representation learning.

- We propose a model-agnostic representation learning framework called RLMRec, which is guided by our theoretical findings. This framework leverages contrastive or generative modeling techniques to enhance the quality of learned representations.

- We establish a theoretical foundation to demonstrate the effectiveness of incorporating textual signals in enhancing the representation learning of existing recommenders. By utilizing mutual information maximization as the optimization direction, we show how textual signals can improve the representation quality.

- We integrate RLMRec with various state-of-the-art recommender models and validate the effectiveness of our method. Additionally, we analyze the robustness of our framework to noise and incomplete data, demonstrating its ability to handle real-world challenges. To ensure reproducibility, we make the source code available at https://anonymous.4open.science/r/RLMRec-EFD1.

## 2 RELATED WORK

**GNN-enhanced Collaborative Filtering**. Collaborative Filtering (CF), which is a fundamental technique in recommendation systems, has been extensively studied over the years [17, 30]. Recently, an emerging research direction involves leveraging historical user-item interactions to construct a bipartite graph and utilizing graph neural networks (GNNs) to capture high-order collaborative relationships. These graph-based methods, such as NGCF [35], GCCF [6], LightGCN [11], have demonstrated state-of-the-art performance, improving recommendation effectiveness. However, the sparsity and noise in implicit feedback data pose challenges to graph-based methods. To address these challenges, researchers have started exploring the use of self-supervised learning (SSL) techniques as auxiliary learning objectives to enhance robustness in recommendations [14, 44]. Among various SSL techniques, contrastive learning has emerged as a prominent solution in collaborative filtering models. Methods like SGL [37], SimGCL [43], NCL [20], LightGCL [3] leverage contrastive data augmentation to improve recommendation performance. In this work, we take a step further by integrating LLMs with existing CF models to effectively align the knowledge and reasoning abilities of LLMs with the collaborative

relation learning for enhancing recommendation performance.

**Large Language Models for Recommendation**. Recently, there has been a growing interest in exploring the use of Large Language Models (LLMs) to benefit recommendation systems [7, 19, 22, 38]. Several studies have leveraged LLMs as inference models by designing prompts that align them with recommendation tasks. For example, P5 [9] converts the user interaction data into textual prompts using item indexes, which are then used for language model training. Chat-REC [8] builds a conversational recommender by transforming user profiles and interactions into prompts for LLMs to generate recommendations. InstructRec [46] and TALL-Rec [1] employ instructional designs to define recommendation tasks and fine-tune LLMs to align with these instructions for generating recommendations. However, using LLMs directly as inference models for recommendation tasks presents challenges, such as high computational costs and slow inference times. These challenges hinder the practical deployment of such models in real-world recommender systems. To address this gap, the proposed approach adopts a theoretically grounded paradigm of mutual information maximization to align the knowledge of LLMs with collaborative relation modeling, enabling scalable and effective recommendations.

## 3 METHODOLOGY

### 3.1 Theoretical Basis of RLMRec

**Collaborative Filtering**. In our recommendation scenario, we have a set of users $\mathcal{U} = u_1, ..., u_I$ and a set of items $\mathcal{V} = v_1, ..., v_J$. The observed user-item interactions are represented by $\mathcal{X}$. In learning-based recommenders, each user and item is assigned initial embeddings $\mathbf{x}_u$ and $\mathbf{x}_v$. The goal is to learn user and item representations $\mathbf{e}_u, \mathbf{e}_v$ through a recommender model (*i.e.*, $\mathbf{e}_u, \mathbf{e}_v = \mathcal{R}(\mathbf{x}_u, \mathbf{x}_v)$) that maximizes the posterior distribution shown below:

$$p(\mathbf{e}|\mathcal{X}) \propto p(\mathcal{X}|\mathbf{e})p(\mathbf{e}). \tag{1}$$

In practical recommendation scenarios, the observed user-item interactions $\mathcal{X}$ often contain noise, including false positives (*e.g.*, misclicks or interactions influenced by popularity bias) and false negatives (*e.g.*, users do not interact with unseen but interested items). As a result, the learned representation $\mathbf{e}$ can also be affected by this noise, which negatively impacts recommendation accuracy. In this work, we introduce a hidden prior belief $\mathbf{z}$ that is inherently beneficial for recommendation. This prior belief helps identify the true positive samples in $\mathcal{X}$. Hence, the generation of representation $\mathbf{e}$ involves a combination of the advantageous prior belief $\mathbf{z}$ and the unavoidable noise present during the learning process.

**Text-enhanced User Preference Learning**. To mitigate the impact of irrelevant signals on the representation, it is necessary to incorporate auxiliary informative cues. One approach is to introduce textual information, *e.g.*, user and item profiles, which provide insights for user preference learning. These profiles can be encoded using language models to generate representations $\mathbf{s} \in \mathbb{R}^{d_s}$ that effectively capture the semantic aspects of user preferences. Importantly, both $\mathbf{s}$ and $\mathbf{e}$ capture shared information that is relevant to the aspects associated with user-item interactions. This shared information is crucial as it indicates the inclusion of beneficial aspects for recommendation, aligning with the prior belief $\mathbf{z}$.

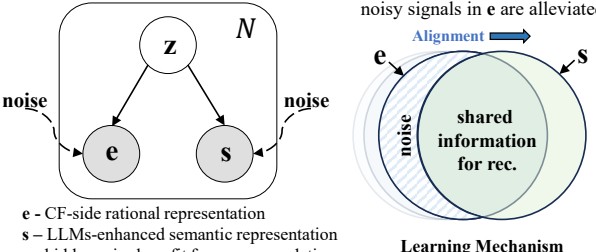

**e** - CF-side rational representation
**s** – LLMs-enhanced semantic representation
**z** - hidden prior benefit for recommendation

**Figure 2: The type of directed graph model under consideration. As the alignment between CF-side representation and LLM-enhanced representation, the noisy effects in the learned representations $e$ are alleviated in RLMRec.**

With the collaborative-side representation $\mathbf{e}$ and textual-side representation $\mathbf{s}$, both of which contain recommendation-beneficial information generated from $\mathbf{z}$, our objective is to learn the optimal value of $\mathbf{e}$ denoted as $\mathbf{e}^*$, by maximizing the conditional probability:

$$\mathbf{e}^* = \arg \max_{\mathbf{e}} \mathbb{E}_{p(\mathbf{e},\mathbf{s})}[p(\mathbf{z}, \mathbf{s}|\mathbf{e})]. \tag{2}$$

The underlying intuition behind maximizing the conditional probability is to ensure that the learnable representation $\mathbf{e}$ from recommender models incorporates purer information generated from the prior belief $\mathbf{z}$ and the shared information with the semantic representation $\mathbf{s}$. By doing so, the relevance and benefits of the learned representations $\mathbf{e}$ for recommendation are enhanced.

**Theorem 1**. *Maximizing the posteriori probability* $\mathbb{E}_{p(\mathbf{e},\mathbf{s})}[p(\mathbf{z}, \mathbf{s}|\mathbf{e})]$ *given the hidden prior belief* $\mathbf{z}$, *is equivalent to maximizing the mutual information* $I(e; s)$ *between the CF-side relational representation* $\mathbf{e}$ *and LLM-side semantic representation* $\mathbf{s}$.

**Proof.** It is important to note that since the profiles of users and items are fixed, the probability $p(\mathbf{s})$ remains constant during the learning process. Therefore, we can deduce the following:

$$\mathbb{E}_{p(\mathbf{e},\mathbf{s})}[p(\mathbf{z}, \mathbf{s}|\mathbf{e})] \propto \mathbb{E}_{p(\mathbf{e},\mathbf{s})} \log[\int_{\mathbf{z}} \frac{p(\mathbf{z}, \mathbf{s}|\mathbf{e})}{p(\mathbf{s})} \, d\mathbf{z}] \tag{3}$$

$$= \mathbb{E}_{p(\mathbf{e},\mathbf{s})} \log[\frac{\int_{\mathbf{z}} p(\mathbf{z}, \mathbf{e}|\mathbf{s}) \, d\mathbf{z}}{p(\mathbf{e})}] \tag{4}$$

$$= \mathbb{E}_{p(\mathbf{e},\mathbf{s})} \log[\frac{p(\mathbf{e}|\mathbf{s})}{p(\mathbf{e})}] = I(\mathbf{e}, \mathbf{s}). \tag{5}$$

Let's consider $\mathbf{e}$ and $\mathbf{s}$ as data samples, assuming that we have $N$ pairwise corresponding elements of $\mathbf{e}$ and $\mathbf{s}$, forming the sets $\mathbf{E} = \{\mathbf{e}_1, \ldots, \mathbf{e}_i, \ldots, \mathbf{e}_N\}$ and $\mathbf{S} = \{\mathbf{s}_1, \ldots, \mathbf{s}_i, \ldots, \mathbf{s}_N\}$, respectively. Based on this, we optimize the mutual information as follows.

**Theorem 2**. *By introducing the density ratio to preserve mutal information* [24] $f(\mathbf{s}, \mathbf{e}) \propto p(\mathbf{s}|\mathbf{e})/p(\mathbf{s})$, *the maximization of* $I(\mathbf{e}_i; \mathbf{s}_i)$ *can be reformulated as maximizing the following lower bound:*

$$\mathbb{E} \log[\frac{f(\mathbf{s}_i, \mathbf{e}_i)}{\sum_{\mathbf{s}_j \in \mathbf{S}} f(\mathbf{s}_j, \mathbf{e}_i)}]. \tag{6}$$

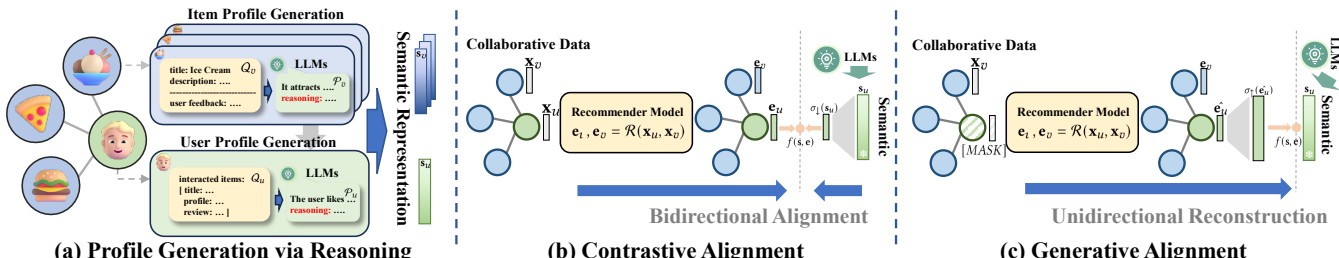

**(a) Profile Generation via Reasoning**          **(b) Contrastive Alignment**          **(c) Generative Alignment**

**Figure 3: The overall framework of our proposed LLM-enhanced representation learning framework RLMRec.**

**Proof.** Based on the property of mutual information, we have $I(\mathbf{e}_i, \mathbf{s}_i) = I(\mathbf{s}_i, \mathbf{e}_i)$. With this in mind, we make the deductions as:

$$I(\mathbf{s}_i, \mathbf{e}_i) \geq I(\mathbf{s}_i, \mathbf{e}_i) - \log(N) = -\mathbb{E} \log[\frac{p(\mathbf{s}_i)}{p(\mathbf{s}_i|\mathbf{e}_i)} N] \tag{7}$$

$$\geq -\mathbb{E} \log[1 + \frac{p(\mathbf{s}_i)}{p(\mathbf{s}_i|\mathbf{e}_i)}(N-1)] \tag{8}$$

$$= -\mathbb{E} \log[1 + \frac{p(\mathbf{s}_i)}{p(\mathbf{s}_i|\mathbf{e}_i)}(N-1)\mathbb{E}_{\mathbf{s}_j \in S_{neg}} \frac{p(\mathbf{s}_j|\mathbf{e}_i)}{p(\mathbf{s}_j)}] \tag{9}$$

$$\approx -\mathbb{E} \log[1 + \frac{p(\mathbf{s}_i)}{p(\mathbf{s}_i|\mathbf{e}_i)} \sum_{\mathbf{s}_j \in S_{neg}} \frac{p(\mathbf{s}_j|\mathbf{e}_i)}{p(\mathbf{s}_j)}] \tag{10}$$

$$= \mathbb{E} \log[\frac{f(\mathbf{s}_i, \mathbf{e}_i)}{\sum_{\mathbf{s}_j \in S} f(\mathbf{s}_j, \mathbf{e}_i)}]. \tag{11}$$

Here, $S_{neg}$ represents the negative samples when considering the $i$-th sample (*i.e.*, $S_{neg} = S \setminus s_i$). Up to this point, we have derived, from a theoretical perspective, how to alleviate noisy effects in representations by introducing external knowledge. However, this approach also presents two challenges: i) **Challenge 1**: How to obtain effective descriptions of users and items that capture their interaction preferences. ii) **Challenge 2**: How to involves effectively modeling the density ratio $f(\mathbf{s}, \mathbf{e})$ to maximize the mutual information between $\mathbf{e}$ and $\mathbf{s}$. In the following sections, we discuss potential solutions to address these two challenges.

## 3.2 User/Item Profiling Paradigm

In our previous derivation, we emphasize the importance of obtaining textual descriptions, referred to as profiles, for users and items. These profiles play a crucial role in mitigating the impact of noise in the learned representations of recommenders and enable a semantic understanding of users' and items' interaction preferences. Ideally, user and item profiles should exhibit the following characteristics:

- **User profile**: should effectively encapsulate the particular types of items that users are inclined to favor, allowing for a comprehensive representation of their personalized tastes and preferences.
- **Item profile**: It should eloquently articulate the specific types of users that the item is apt to attract, providing a clear representation of the item's characteristics and qualities that align with the preferences and interests of those users.

In some cases, the original data may include textual properties related to users and items. For example, in the Yelp dataset, users provide reviews for visited businesses, and businesses have attributes such as location and category. However, such textual data often contains extraneous noise, leading to common predicaments:

i) **Missing Attributes**: Some attributes of certain items or users may be missing; ii) **Noisy Textual Data**: The text itself may be contaminated with a plethora of noise that is irrelevant to users' preferences. For instance, in the Steam dataset, user reviews for games may contain numerous special symbols or irrelevant information. These challenges make it difficult to distill useful user and item profiles from text. As a result, prevailing models often convert low-noise attributes into one-hot encodings without effectively leveraging the semantic information present in the textual data.

Fortunately, recent advancements in Large Language Models (LLMs) have unleashed their remarkable text processing capabilities, enabling them to address a wide range of NLP tasks, including text denoising and summarization. This pivotal development opens up new possibilities for generating user and item profiles from the noisy textual features inherent in the dataset. Leveraging the tremendous potential of LLMs, we propose a paradigm for profile generation that capitalizes on collaborative information. Considering that datasets often contain a higher proportion of textual descriptions for item attributes compared to user attributes, our approach takes an item-to-user perspective, as outlined below.

### 3.2.1 Profile Generation via Reasoning.
Recent research has demonstrated the effectiveness of incorporating process reasoning in LLMs to mitigate hallucination and improve the quality of generated outputs. Building upon these findings, we have meticulously designed the system prompt $\mathcal{S}_{u/v}$ as part of the input provided to LLMs. The objective is to clearly define its functionality in generating user profile for user $u$ or item profile for item $v$ by precisely specifying the input-output content and desired output format. Importantly, we explicitly emphasize the inclusion of reasoning processes as an integral part of the generated output. By combining this system prompt with user/item profile generation prompts $Q_u$ and $Q_v$, we can leverage LLMs to generate accurate profiles. The specific process is outlined as follows:

$$\mathcal{P}_u = LLMs(\mathcal{S}_u, Q_u), \quad \mathcal{P}_v = LLMs(\mathcal{S}_v, Q_v) \tag{12}$$

### 3.2.2 Item Prompt Construction.
we categorize the textual information of an item $v \in \mathcal{V}$ into four types: title $\alpha$, original description $\beta$, dataset-specific attributes $\gamma = \gamma_1, ..., \gamma_{|\gamma|}$, and a collection of $n$ reviews from users $\mathbf{r} = r_1, ..., r_n$. Based on these categories, we can formally outline the arrangement of the input prompt $Q_v$ for item-profile generation as follows:

$$Q_v = f_v(\mathbf{x}) \quad w.r.t. \quad \mathbf{x} = \begin{cases} [\alpha, \beta], & \text{if } \beta \text{ exists,} \\ [\alpha, \gamma, \hat{\mathbf{r}} \subset \mathbf{r}], & \text{other wise.} \end{cases} \tag{13}$$

In our approach, we use a function $f_v(\cdot)$ specific to each item, which combines various text features into a single string. If the original description $\beta$ is missing, we randomly sample a subset of reviews $\hat{\mathbf{r}}$ and combine them with the attributes for input. By incorporating item descriptions or user reviews, our prompts provide precise information to Large Language Models, ensuring that the generated item profiles accurately reflect appealing characteristics.

*3.2.3* **User Prompt Construction.** To generate the profile of user $u$, we leverage collaborative information, assuming that we have already generated the item profiles beforehand. Specifically, we consider the items interacted with by user $u$ as $\mathcal{I}_u$ and uniformly sample a subset of items $\hat{\mathcal{I}}_u \subset \mathcal{I}_u$. For each item $v$ in $\hat{\mathcal{I}}_u$, we concatenate its textual attributes as $\mathbf{c}_v = [\alpha, \mathcal{P}_v, r_u^v]$, where $r_u^v$ represents the review provided by user $u$. The input prompt $Q_u$ for user-profile generation can be defined as follows:

$$Q_u = f_u(\{\mathbf{c}_v | v \in \hat{\mathcal{I}}_u\}). \quad (14)$$

The function $f_u(\cdot)$ serves a similar purpose to $f_v(\cdot)$ by organizing the textual content into a coherent string. Each textual attribute $\mathbf{c}_v$ includes user reviews, which authentically reflect their genuine opinions. This construction of the user prompt provides valuable insights into their true preferences. Due to space constraints, we have included the detailed design of the prompt, including $\mathcal{S}$, $Q$, and $f_{u/v}(\cdot)$, along with sample examples in Appendix A.2.

## 3.3 Density Ratio Modeling for Mutual Information Maximization

In this section, we outline the process of modeling the density ratio, denoted as $f(\mathbf{s}_i, \mathbf{e}_i)$, with the objective of maximizing the mutual information $I(\mathbf{s}_i, \mathbf{e}_i)$. First of all, it is important to note that we have previously generated user/item profiles $\mathcal{P}_{u/v}$ that showcase their interaction preferences. As such, it is logical to encode the semantic representation $\mathbf{s}$ based on these profiles as follow:

$$\mathbf{s}_u = \mathcal{T}(\mathcal{P}_u), \quad \mathbf{s}_v = \mathcal{T}(\mathcal{P}_v). \quad (15)$$

Here, $\mathcal{T}(\cdot)$ refers to a cutting-edge technology known as a text embedding model [15, 29], which has been shown to effectively transform diverse text inputs into fixed-length vectors that retain their inherent meaning and contextual information.

According to [24], the density ratio $f(\mathbf{s}_i, \mathbf{e}_i)$ can be interpreted as a positive real-valued score measurement function that captures the similarity between $\mathbf{s}_i$ and $\mathbf{e}_i$. A more accurate modeling of the density ratio [28] can have a positive impact on the alignment between CF-side rational representations and LLMs-enhanced semantic representations, helping to mitigate the influence of noisy signals in representation learning. In this context, we propose two types of modeling approaches that are well-suited for achieving this alignment. The first approach is contrastive modeling, which has been extensively validated [16, 37] for effectively aligning different views bidirectionally, such as through pull and push pairs. The second approach is mask-reconstruction generative modeling, which is widely used as a self-supervised mechanism for reconstructing the partially masked input from data itself [10, 13]. By employing CF-side representations to reconstruct the semantic representations, we can effectively align these two forms of information.

*3.3.1* **Contrastive Alignment.** As depicted in Fig 3 (b), we denote the specific implementation of $f(\mathbf{s}_i, \mathbf{e}_i)$ as contrastive alignment.

$$f(\mathbf{s}_i, \mathbf{e}_i) = exp(sim(\sigma_{\downarrow}(\mathbf{s}_i), \mathbf{e}_i)). \quad (16)$$

The function $sim(\cdot)$ represents the cosine similarity, while $\sigma_{\downarrow}$ denotes a multi-layer perception that maps the semantic representation $\mathbf{s}_i$ into the feature space of $\mathbf{e}_i$. In our contrastive alignment, we treat $\mathbf{e}_i$ and $\mathbf{s}_i$ as positive sample pairs. During the learning process, these pairs are pulled towards each other to align their representations. In the specific implementation, the objective is to bring positive sample pairs closer within a batch while considering the remaining samples as negatives.

*3.3.2* **Generative Alignment.** Taking inspiration from recent research on the masked autoencoder (MAE), which is considered a paradigm of generative self-supervised learning, we propose an additional modeling approach for the density ratio within the MAE.

$$f(\mathbf{s}_i, \mathbf{e}_i) = exp(sim(\mathbf{s}_i, \sigma_{\uparrow}(\hat{\mathbf{e}_u}))) \quad w.r.t. \quad \hat{\mathbf{e}}_i = \mathcal{R}(\{\mathbf{x}\} \setminus \mathbf{x}_i). \quad (17)$$

We employ $\sigma_{\uparrow}$ as a multi-layer perception model to map the representations to the semantic feature space. $\mathbf{x} \setminus \mathbf{x}_i$ represents the initial embedding of the $i$-th sample with masking applied. The generative process follows a single-direction reconstruction approach, focusing on reconstructing the semantic representations exclusively for the masked samples. Specifically, the masking operation involves replacing the initial embedding with a designated mask token (*i.e.*, $[MASK]$), and a random subset of users/items is masked and subsequently reconstructed. This allows us to explore the reconstruction capabilities within the semantic feature space.

With our contrastive and generative alignment method, we effectively align the knowledge of the LLM with the domain of understanding user preferences. This is achieved by combining id-based collaborative relational signals with text-based behavior semantics. We have given the names **RLMRec-Con** and **RLMRec-Gen** to our two proposed modeling approaches, respectively. In our experiments conducted on real-world data, we will comprehensively evaluate the performance of these two models across various tasks, each showcasing its unique advantages and disadvantages.

## 3.4 Model-agnostic Learning

Up until this point, our focus has been on optimizing the CF-side relational representation $\mathbf{e}$ and LLM-side semantic representation $\mathbf{s}$. Any model that can perform representation learning for users/items can undergo the optimization process described earlier. Hence, our approach is model-agnostic and can seamlessly enhance existing collaborative filtering recommenders. Assuming that the optimization objective of the recommender $\mathcal{R}$ is denoted as $\mathcal{L}_{\mathcal{R}}$, our overall optimization function $\mathcal{L}$ can be formulated as follows:

$$\mathcal{L} = \mathcal{L}_{\mathcal{R}} + \mathcal{L}_{info} \quad w.r.t. \quad \mathcal{L}_{info} = -\mathbb{E} \log[\frac{f(\mathbf{s}_i, \mathbf{e}_i)}{\sum_{\mathbf{s}_j \in S} f(\mathbf{s}_j, \mathbf{e}_i)}], \quad (18)$$

Minimizing the overall optimization function $\mathcal{L}$ corresponds to maximizing the mutual information mentioned earlier.

## 4 EVALUATION

This section presents the experimental evaluation of our RLMRec on multiple datasets to address the following research questions:

**Table 1: Recommendation performance Imprvement of all backbone methods on different datasets in terms of Recall and NDCG. The superscript * indicates the Imprvement is statistically significant where the p-value is less than 0.05.**

| Data | | Amazon-book | | | | | | Yelp | | | | | | Steam | | | | | |
|---|---|---|---|---|---|---|---|---|---|---|---|---|---|---|---|---|---|---|---|
| Backbone | Variants | R@5 | R@10 | R@20 | N@5 | N@10 | N@20 | R@5 | R@10 | R@20 | N@5 | N@10 | N@20 | R@5 | R@10 | R@20 | N@5 | N@10 | N@20 |
| Semantic Embeddings Only | | 0.0081 | 0.0125 | 0.0199 | 0.0072 | 0.0088 | 0.0112 | 0.0013 | 0.0022 | 0.0047 | 0.0014 | 0.0018 | 0.0026 | 0.0033 | 0.0062 | 0.0120 | 0.0031 | 0.0043 | 0.0064 |
| GCCF | Base | 0.0537 | 0.0872 | 0.1343 | 0.0537 | 0.0653 | 0.0807 | 0.0390 | 0.0652 | 0.1084 | 0.0451 | 0.0534 | 0.0680 | 0.0500 | 0.0826 | 0.1313 | 0.0556 | 0.0665 | 0.0830 |
| | RLMRec-Con | **0.0561*** | **0.0899*** | **0.1395*** | **0.0562*** | **0.0679*** | **0.0842*** | **0.0409*** | **0.0685*** | **0.1144*** | **0.0474*** | **0.0562*** | **0.0719*** | **0.0538*** | **0.0883*** | **0.1398*** | **0.0597*** | **0.0713*** | **0.0888*** |
| | RLMRec-Gen | 0.0551* | 0.0891* | 0.1372* | 0.0559* | 0.0675* | 0.0832* | 0.0393 | 0.0654 | 0.1074 | 0.0454 | 0.0535 | 0.0678 | 0.0532* | 0.0874* | 0.1385* | 0.0588* | 0.0702* | 0.0875* |
| | **Best Imprv.** | ↑4.28% | ↑3.10% | ↑3.87% | ↑4.66% | ↑3.98% | ↑4.34% | ↑4.87% | ↑5.06% | ↑5.54% | ↑5.10% | ↑5.24% | ↑5.74% | ↑7.60% | ↑6.90% | ↑6.47% | ↑7.37% | ↑7.22% | ↑6.99% |
| LightGCN | Base | 0.0570 | 0.0915 | 0.1411 | 0.0574 | 0.0694 | 0.0856 | 0.0421 | 0.0706 | 0.1157 | 0.0491 | 0.0580 | 0.0733 | 0.0518 | 0.0852 | 0.1348 | 0.0575 | 0.0687 | 0.0855 |
| | RLMRec-Con | **0.0608*** | **0.0969*** | **0.1483*** | **0.0606*** | **0.0734*** | **0.0903*** | **0.0445*** | **0.0754*** | **0.1230*** | **0.0518*** | **0.0614*** | **0.0776*** | 0.0548* | 0.0895* | 0.1421* | **0.0608*** | 0.0724* | 0.0902* |
| | RLMRec-Gen | 0.0596* | 0.0948* | 0.1446* | 0.0605* | 0.0724* | 0.0887* | 0.0435* | 0.0734* | 0.1209* | 0.0505 | 0.0600* | 0.0761* | **0.0550*** | **0.0907*** | **0.1433*** | 0.0607* | **0.0729*** | **0.0907*** |
| | **Best Imprv.** | ↑6.67% | ↑5.90% | ↑5.10% | ↑5.57% | ↑5.76% | ↑5.49% | ↑5.70% | ↑6.80% | ↑6.31% | ↑5.50% | ↑5.86% | ↑5.87% | ↑6.18% | ↑6.46% | ↑6.31% | ↑5.74% | ↑6.11% | ↑6.08% |
| SGL | Base | 0.0637 | 0.0994 | 0.1473 | 0.0632 | 0.0756 | 0.0913 | 0.0432 | 0.0722 | 0.1197 | 0.0501 | 0.0592 | 0.0753 | 0.0565 | 0.0919 | 0.1444 | 0.0618 | 0.0738 | 0.0917 |
| | RLMRec-Con | **0.0655*** | **0.1017*** | 0.1528* | **0.0652*** | **0.0778*** | 0.0945* | 0.0452* | 0.0763* | 0.1248* | 0.0530* | 0.0626* | 0.0790* | **0.0589*** | **0.0956*** | **0.1489*** | **0.0645*** | **0.0768*** | **0.0950*** |
| | RLMRec-Gen | 0.0644 | 0.1015 | **0.1537*** | 0.0648* | 0.0777* | **0.0947*** | **0.0467*** | **0.0771*** | **0.1263*** | **0.0537*** | **0.0631*** | **0.0798*** | 0.0574* | 0.0940* | 0.1476* | 0.0629* | 0.0752* | 0.0934* |
| | **Best Imprv.** | ↑2.83% | ↑2.31% | ↑4.34% | ↑3.16% | ↑2.91% | ↑3.72% | ↑8.10% | ↑6.79% | ↑5.51% | ↑7.19% | ↑6.59% | ↑5.98% | ↑5.20% | ↑4.03% | ↑3.12% | ↑4.37% | ↑4.07% | ↑3.60% |
| SimGCL | Base | 0.0618 | 0.0992 | 0.1512 | 0.0619 | 0.0749 | 0.0919 | 0.0467 | 0.0772 | 0.1254 | 0.0546 | 0.0638 | 0.0801 | 0.0564 | 0.0918 | 0.1436 | 0.0618 | 0.0738 | 0.0915 |
| | RLMRec-Con | **0.0633*** | **0.1011*** | **0.1552*** | **0.0633*** | **0.0765*** | **0.0942*** | **0.0470*** | **0.0784*** | **0.1292*** | 0.0546 | 0.0642 | **0.0814*** | **0.0582*** | **0.0945*** | **0.1482*** | **0.0638*** | **0.0760*** | **0.0942*** |
| | RLMRec-Gen | 0.0617 | 0.0991 | 0.1524* | 0.0622 | 0.0752 | 0.0925* | 0.0464 | 0.0767 | 0.1267 | 0.0541 | 0.0634 | 0.0803 | 0.0572 | 0.0929 | 0.1456* | 0.0627 | 0.0747* | 0.0926* |
| | **Best Imprv.** | ↑2.43% | ↑1.92% | ↑2.65% | ↑2.26% | ↑2.14% | ↑2.50% | ↑0.64% | ↑1.55% | ↑3.03% | − | ↑0.63% | ↑1.62% | ↑3.19% | ↑2.94% | ↑1.53% | ↑3.24% | ↑2.98% | ↑2.95% |
| DCCF | Base | 0.0662 | 0.1019 | 0.1517 | 0.0658 | 0.0780 | 0.0943 | 0.0468 | 0.0778 | 0.1249 | 0.0543 | 0.0640 | 0.0800 | 0.0561 | 0.0915 | 0.1437 | 0.0618 | 0.0736 | 0.0914 |
| | RLMRec-Con | 0.0665 | 0.1040* | **0.1563*** | 0.0668 | 0.0798* | 0.0968* | **0.0486*** | **0.0813*** | **0.1321*** | **0.0561*** | **0.0663*** | **0.0836*** | **0.0572*** | **0.0929*** | **0.1459*** | **0.0627*** | **0.0747*** | **0.0927*** |
| | RLMRec-Gen | **0.0666*** | **0.1046*** | 0.1559* | **0.0670*** | **0.0801*** | **0.0969*** | 0.0475 | 0.0785 | 0.1281* | 0.0549 | 0.0646 | 0.0815 | 0.0570* | 0.0918 | 0.1430 | 0.0625 | 0.0741 | 0.0915 |
| | **Best Imprv.** | ↑0.60% | ↑2.65% | ↑3.03% | ↑1.82% | ↑2.69% | ↑2.76% | ↑3.85% | ↑4.50% | ↑5.76% | ↑3.31% | ↑3.59% | ↑4.50% | ↑2.14% | ↑1.53% | ↑1.53% | ↑1.46% | ↑1.49% | ↑1.42% |
| AutoCF | Base | 0.0689 | 0.1055 | 0.1536 | 0.0705 | 0.0828 | 0.0984 | 0.0469 | 0.0789 | 0.1280 | 0.0547 | 0.0647 | 0.0813 | 0.0519 | 0.0853 | 0.1358 | 0.0572 | 0.0684 | 0.0855 |
| | RLMRec-Con | **0.0695*** | **0.1083*** | **0.1586*** | 0.0704 | **0.0837*** | **0.1001*** | 0.0488* | 0.0814* | 0.1319* | 0.0562* | 0.0663* | 0.0835* | **0.0540*** | 0.0876* | 0.1372* | 0.0593* | 0.0704* | 0.0872* |
| | RLMRec-Gen | 0.0693 | 0.1069* | 0.1581* | 0.0701 | 0.0830 | 0.0996 | **0.0493*** | **0.0828*** | **0.1330*** | **0.0572*** | **0.0677*** | **0.0848*** | 0.0539* | **0.0888*** | **0.1410*** | **0.0593*** | **0.0710*** | **0.0886*** |
| | **Best Imprv.** | ↑0.87% | ↑2.65% | ↑3.26% | ↓0.14% | ↑1.87% | ↑1.73% | ↑5.12% | ↑4.94% | ↑3.91% | ↑4.57% | ↑4.64% | ↑4.31% | ↑4.05% | ↑4.10% | ↑3.83% | ↑3.67% | ↑3.80% | ↑3.63% |

- **RQ1**: Does our proposed RLMRec improve upon existing state-of-the-art recommenders across various experimental settings?
- **RQ2**: Do the LLM-enhanced semantic representations contribute to the recommendation performance improvement?
- **RQ3**: Does our proposed framework effectively tackle the issue of noisy data through cross-view semantic alignment?
- **RQ4**: What is the potential of our model as a pre-training framework for enhancing the performance of recommender systems?
- **RQ5**: How does our RLMRec perform *w.r.t* training efficiency?

## 4.1 Experimental Settings

*4.1.1 **Datasets**.* We conduct evaluations of our RLMRec on three public datasets: **Amazon-book**: This dataset contains user ratings and corresponding reviews for books sold on Amazon. **Yelp**: This dataset is a user-business dataset that provides extensive textual category information about various businesses. **Steam**: This dataset consists of textual feedback given by users for electronic games available on the Steam platform. Following the similar settings in [35, 42, 43] for data preprocessing, we filter out interactions with ratings below 2 for the Amazon-book and below 3 for the Yelp. No filtering is applied to the Steam dataset due to the absence of rating scores. We then perform k-core filtering and divided each dataset into training, validation, and testing sets using a 3:1:1 ratio. Please refer to Table 5 in Appendix for a summary of the dataset statistics.

*4.1.2 **Evaluation Protocols and Metrics**.* To ensure comprehensive evaluation and mitigate bias, we adopt the all-rank protocol [11, 36, 37] across all items to accurately assess our recommendations. We use two widely adopted ranking-based metrics: Recall@N and NDCG@N, which measure the model effectiveness.

*4.1.3 **Base Models**.* We evaluate the effectiveness of our RLMRec by seamlessly integrating it with state-of-the-art representation-based recommenders. This approach allows us to assess its model-agnostic performance gain in comparison to base models.

**Table 2: Comparison with LLMs-enhanced Approaches.**

| Data | | Amazon-book | | Yelp | |
|---|---|---|---|---|---|
| Backb. | Variants | R@20 | N@20 | R@20 | N@20 |
| Light-GCN | Base | 0.1411 | 0.0856 | 0.1157 | 0.0733 |
| | KAR | 0.1416+0.3% | 0.0863+0.8% | 0.1194+3.2% | 0.0756+3.1% |
| | RLMRec-Con | 0.1483+5.1% | 0.0903+5.5% | 0.1230+6.3% | 0.0776+5.9% |
| | RLMRec-Gen | 0.1446+2.5% | 0.0887+3.6% | 0.1209+4.5% | 0.0761+3.8% |
| SGL | Base | 0.1473 | 0.0913 | 0.1197 | 0.0753 |
| | KAR | 0.1372−6.9% | 0.0875−4.2% | 0.1208+0.9% | 0.0761+1.1% |
| | RLMRec-Con | 0.1528+3.7% | 0.0945+3.5% | 0.1248+4.3% | 0.0790+4.9% |
| | RLMRec-Gen | 0.1537+4.3% | 0.0947+3.7% | 0.1263+5.5% | 0.0798+6.0% |

- **GCCF** [6]: It simplifies graph-based recommender design by re-evaluating the role of non-linear operations in GNNs.
- **LightGCN** [11]: It creates a lightweight recommender by streamlining redundant neural modules in graph message passing.
- **SGL** [37]: It utilizes node/edge dropout as a data augmentator to generate diverse perspectives for contrastive learning.
- **SimGCL** [43]: It enhances recommendation performance by introducing an augmentation-free view generation technique.
- **DCCF** [26]: It captures intent-wise relationships for recommendation purposes using disentangled contrastive learning.
- **AutoCF** [41]: It is a self-supervised masked autoencoder to automate the process of data augmentation for recommendation.

*4.1.4 **Implementation Details**.* To ensure fair comparison, all the baseline models are evaluated using their released implementation source codes. The dimension of representations (*i.e.,* **x** and **e**) is set to 32 for all base models. We determine the hyperparameters specific to each model through grid search under various settings. To generate user and item profiles, we leverage the ChatGPT model (specifically, gpt-3.5-turbo) provided by OpenAI. We use the text-embedding-ada-002 [23] to generate semantic representations **s**. During training, all methods are trained with a fixed batch size of 4096 and a learning rate of 1e-3 using the Adam optimizer. We

adopt the early stop technique based on the model's performance on the validation set. For detailed implementation of our RLMRec, we provide the source code for reference.

## 4.2 Performance Comparison (RQ1)

**Model-agnostic Performance Gain**. To demonstrate the effectiveness of RLMRec in improving recommendation performance, we integrate it into six state-of-the-art collaborative filtering models. We conduct experiments using 5 random initializations and report the average results in Table 1. The evaluation results reveal several interesting observations, as outlined below:

- Overall, we consistently observe that integrating RLMRec with the backbone recommenders leads to improved performance compared to the original versions. This provides compelling evidence for the effectiveness of RLMRec. We attribute these improvements to two key factors: i) RLMRec enables the accurate user/item profiling paradigm empowered by LLMs, enhancing the representation of rich semantic information derived from user interaction behaviors. ii) Through our cross-view mutual information maximization, the CF-side relational embeddings and LLM-side semantic representations work cooperatively to enhance each other. This collaborative effort effectively filters out irrelevant noise in the recommendation features.

- It is clear that both contrastive and generative modeling approaches generally improve performance. However, it is important to note that the contrastive approach exhibits superior performance when combined with various backbones like GCCF and SimGCL. Conversely, when applied to AutoCF, which involves masked reconstruction, RLMRec-Gen shows more significant improvements. We speculate that the mask operation functions as a form of regularization, leading to better results when used in conjunction with methods that employ a generative approach.

**Superiority over LLM-enhanced Approach**. In addition, we conduct a comparative evaluation of the effectiveness of RLMRec in comparison to KAR [40], a recent LLM-enhanced user behavior modeling approach. KAR aims to generate textual user/item descriptions to enhance the learning of user preferences for the CTR task. To ensure a fair comparison, we utilized the same semantic representation as in our approach and employed two classic methods (LightGCN and SGL) as the backbone models. This could be attributed to the fact that, while KAR incorporates textual information into the learning of user preferences, it treats the semantic representation merely as input features for the model. As a result, it may not effectively align the textual knowledge with the user behavior representations and could be more susceptible to irrelevant noise from either user behaviors or the LLM knowledge base.

## 4.3 Ablation Study (RQ2)

In this section, we examine the impact of integrating semantic representations on performance. To do this, we shuffle the acquired semantic representations, creating a misalignment with collaborative relational representation and LLM's knowledge. We use the default semantic encoding model, text-embedding-ada-002 [2], and also experiment with advanced models like Contriever [15] and

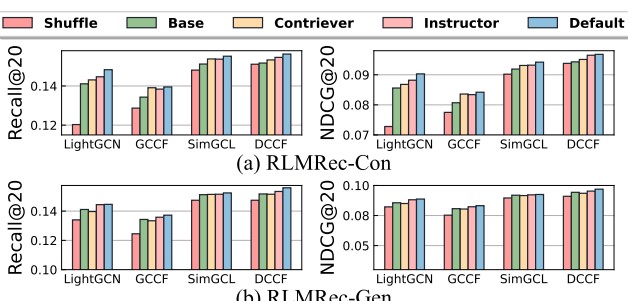

**Figure 4: Ablation study on variant text embedding models conducted on the Amazon-book dataset. Shuffling involves reordering user/item embeddings.**

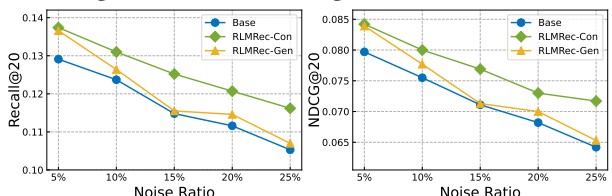

**Figure 5: Comparing performance on different noise ratios in the Amazon-book dataset with LightGCN as the base model.**

Instructor [29]. We evaluate our approach on four backbone methods (*i.e.*, LightGCN, GCCF, SimGCL, and DCCF). The results are summarized in Figure 4, leading to two key observations.

- After randomly rearranging the semantic representations to disrupt the correlation between collaborative and semantic signals, we observe a decrease in performance for both RLMRec-Con and RLMRec-Gen on the evaluated backbone models compared to their original performance. This suggests that the shuffled representations introduce noise due to the mismatch between semantic and collaborative information. It provides evidence that accurate alignment between the semantic knowledge of the LLM and collaborative relationships among users is crucial for enhancing recommendation performance.

- When we utilize variant text embedding models like Contriever and Instructor, our RLMRec still enhances the base performance, similar to the default setting with text-embedding-ada-002. This indicates that our RLMRec can effectively leverage an appropriate text encoder capable of transferring textual semantics into preference representations to improve the performance of the recommender backbone. Moreover, the ability of text embedding models to capture semantic information with higher accuracy can lead to even more significant improvements.

## 4.4 In-depth Analysis of RLMRec (RQ3 – RQ5)

*4.4.1* *Performance w.r.t. Noisy Data (RQ3)*. We assess the robustness of RLMRec to data noise by adding non-existent interactions to the original training data. Noise levels range from 5% to 25% relative to the training set size. Using the Amazon dataset, we compare the performance of vanilla LightGCN with LightGCN enhanced by our RLMRec-Con/Gen. Key findings from Fig 5 are:

- (i) Both RLMRec-Con and RLMRec-Gen consistently outperform the LightGCN backbone model at all noise levels. This highlights

**Table 3: Performance comparison with different initialized parameters from various pre-training methods on the Yelp.**

| Metric | Recall | | | NDCG | | |
|---|---|---|---|---|---|---|
| Pretrained Params | @5 | @10 | @20 | @5 | @10 | @20 |
| None | 0.0274 | 0.0462 | 0.0820 | 0.0203 | 0.0270 | 0.0375 |
| Base | 0.0304 | 0.0557 | 0.0971 | 0.0229 | 0.0319 | 0.0439 |
| RLMRec-Con | 0.0359 | 0.0613 | 0.1034 | 0.0261 | 0.0352 | 0.0475 |
| RLMRec-Gen | **0.0362** | 0.0612 | **0.1068** | **0.0263** | **0.0353** | **0.0484** |

**Table 4: RLMRec's Efficiency with Various Recommenders.**

| Ama-Variants | GCCF | LightGCN | SGL | SimGCL | DCCF | AutoCF |
|---|---|---|---|---|---|---|
| Base | 0.88s | 1.01s | 2.18s | 2.62s | 2.26s | 2.73s |
| RLMRec-Con | 1.95s | 1.94s | 2.58s | 3.02s | 2.49s | 2.96s |
| RLMRec-Gen | 1.72s | 1.76s | 2.36s | 2.69s | 2.29s | 2.96s |
| **Yelp-Variants** | **GCCF** | **LightGCN** | **SGL** | **SimGCL** | **DCCF** | **AutoCF** |
| Base | 1.11s | 1.26s | 2.80s | 3.35s | 3.02s | 3.96s |
| RLMRec-Con | 2.39s | 2.57s | 3.27s | 3.95s | 3.42s | 4.41s |
| RLMRec-Gen | 2.03s | 2.12s | 3.20s | 3.50s | 3.24s | 4.39s |
| **Steam-Variants** | **GCCF** | **LightGCN** | **SGL** | **SimGCL** | **DCCF** | **AutoCF** |
| Base | 2.05s | 2.27s | 5.42s | 6.47s | 9.31s | 8.44s |
| RLMRec-Con | 4.32s | 4.67s | 6.77s | 7.88s | 10.18s | 10.06s |
| RLMRec-Gen | 3.33s | 3.81s | 6.10s | 6.89s | 9.57s | 9.89s |

the advantages of incorporating semantic information and leveraging mutual information to filter out irrelevant data, resulting in improved recommendations and robustness over noise.

- (ii) RLMRec-Con has shown better resistance to data noise compared to RLMRec-Gen. This is likely due to the inherent noise introduced by the generative method through node masking. In contrast, contrastive methods encounter less noise, leading to superior performance under the same noise ratio.

*4.4.2 Performance in Pre-training Scenarios (RQ4).* We investigate the potential of our semantically involved training mechanism as a pre-training technique to provide informative representations as initial embeddings for downstream models. Using the Yelp dataset, we utilize data from 2012 to 2017 for pre-training and divide the data from 2018 to 2019 into a training set, a validation set, and a test set (the downstream dataset). Both datasets contain the same users and items. We train vanilla LightGCN and our model on the pre-training dataset. The learned parameters are used to initialize the embeddings for vanilla LightGCN, which is then trained on the downstream dataset. Key findings from Table 3 are:

- Pre-training with parameters yields superior results compared to no pre-training, regardless of whether it was done with the base model or our RLMRec. This suggests that the pre-training dataset contains valuable collaborative information that helps predict user/item preferences and benefits downstream tasks.
- Both RLMRec-Con and RLMRec-Gen provide better pre-training benefits compared to pre-training with the base model alone, with RLMRec-Gen achieving the best results. This highlights the advantage of incorporating semantic information and the effectiveness of generative methods in pre-training scenarios, potentially due to the regulatory function of the mask operation, preventing overfitting on the pre-training dataset.

*4.4.3 Analysis of Training Efficiency (RQ5).* We analyze the time complexity of using RLMRec. The theoretical time complexity of the multi-layer perception ($\sigma_\uparrow$ and $\sigma_\downarrow$) for both RLMRec-Con

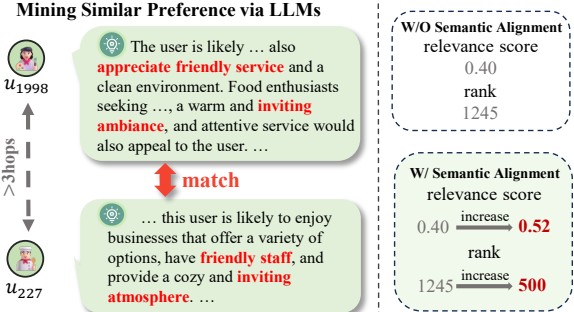

**Figure 6: Case study on capturing global user dependencies.**

and RLMRec-Gen is $O(N \times d_s \times d_e)$. For RLMRec-Con, the loss computation introduces an additional complexity of $O(N^2 \times d)$. For RLMRec-Gen, the time complexity is $O(M \times d + M \times N \times d)$, where the masking operation accounts for $M \times d$, with $M$ representing the number of masked nodes. In Table 4, we present the epoch time of training on a server with an Intel Xeon Silver 4314 CPU and an NVIDIA RTX 3090 GPU. The results show that the time cost of RLMRec-Gen is consistently lower than that of RLMRec-Con. This is primarily because the value of $N$ in RLMRec-Con is determined by the batch size, which tends to be larger than the number of masked nodes M in RLMRec-Gen. Additionally, for larger models with improved performance, the additional time complexity is only around 10% to 20% compared to the original time.

### 4.5 Case Study

In our investigation, we explore the integration of LLM-enhanced semantics to capture global user relationships that are not easily captured through direct message passing on the graph. Figure 6 presents a case study where the distance between user $u_{1998}$ and $u_{227}$ exceeds 3 hops. To evaluate the models' ability to capture their relationship, we examine the similarity of user representations. We compared LightGCN and RLMRec-Con, both using the same backbone. Two metrics were introduced: a relevance score for user $u_{1998}$ and the ranking of its long-distance neighbors (greater than 3 hops) based on the score. By incorporating semantic information, derived from language models that highlight shared interests between $u_{1998}$ and $u_{227}$ (e.g., their preference for friendly service), both the relevance score and ranking increased. This indicates that the learned representations from RLMRec effectively capture global collaborative relationships beyond ID-based recommendation techniques.

## 5 CONCLUSION

This paper presents RLMRec, a model-agnostic framework that leverages Large Language Models (LLMs) to improve the representation performance of recommender systems. We introduce a collaborative profile generation paradigm and a reasoning-driven system prompt, emphasizing the inclusion of reasoning processes in the generated output. RLMRec utilizes contrastive and generative alignment techniques to align CF-side relational embeddings with LLM-side semantic representations, effectively reducing feature noise. The framework combines the strengths of general recommenders and LLMs, supported by robust theoretical guarantees, and is extensively evaluated on real-world datasets. Our future investigations will focus on advancing LLM-based reasoning results in recommender systems by providing more insightful explanations.

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

**Table 5: Statistics of the experimental datasets.**

| Dataset | #Users | #Items | #Interactions | Density |
|---------|--------|--------|---------------|---------|
| Amazon-book | 11,000 | 9,332 | 120,464 | $1.2e^{-3}$ |
| Yelp | 11,091 | 11,010 | 166,620 | $1.4e^{-3}$ |
| Steam | 23,310 | 5,237 | 316,190 | $2.6e^{-3}$ |

---

**Algorithm 1:** Training Procedure in RLMRec-Con

**input** : Base model $\mathcal{R}$, implicit feedback $\mathcal{X}$, semantic representation $\mathbf{s}$ for each user & item and learning rate $\eta$

**Result:** Trained model parameters $\Theta$

1 **repeat**
2      uniformly sample batch data $\mathcal{B} = \{(u, v_{pos}, v_{neg})\} \in \mathcal{X}$;
3      inference collaborative-side representation $\mathbf{e}_{u/v}$ with $\mathcal{R}$;
4      calculate model optimization objective $\mathcal{L}_{\mathcal{R}}$ based on $\mathcal{B}$;
5      calculate $L_{info}$ *w.r.t.* Eq (16 & 18) for all $u/v$ in $\mathcal{B}$;
6      $\mathcal{L} = \mathcal{L}_{\mathcal{R}} + \mathcal{L}_{info}$;
7      $\Theta \leftarrow \Theta - \eta\nabla_{\Theta}\mathcal{L}$;
8 **until** *convergence*;

---

**Algorithm 2:** Training Procedure in RLMRec-Gen

**input** : Base model $\mathcal{R}$, implicit feedback $\mathcal{X}$, semantic representation $\mathbf{s}$ for each user & item, learning rate $\eta$ and masking ratio $\alpha$

**Result:** Trained model parameters $\Theta$

1 **repeat**
2      uniformly sample batch data $\mathcal{B} = \{(u, v_{pos}, v_{neg})\} \in \mathcal{X}$;
3      randomly sample a subset of users & items with ratio $\alpha$;
4      replace initial embeddings of masked $u/v$ with $[MASK]$;
5      inference collaborative-side representation $\mathbf{e}_{u/v}$ with $\mathcal{R}$;
6      calculate model optimization objective $\mathcal{L}_{\mathcal{R}}$ based on $\mathcal{B}$;
7      calculate $L_{info}$ *w.r.t.* Eq (17 & 18) for masked $u/v$ in $\mathcal{B}$;
8      $\mathcal{L} = \mathcal{L}_{\mathcal{R}} + \mathcal{L}_{info}$;
9      $\Theta \leftarrow \Theta - \eta\nabla_{\Theta}\mathcal{L}$;
10 **until** *convergence*;

---

# A SUPPLEMENTARY MATERIAL

In the supplementary materials, we provide the training procedure of our proposed framework, RLMRec, through pseudocode. We also offer detailed insights into the design of prompts, accompanied by examples, to visualize the profile generation process within our item-to-user generation paradigm. Finally, we present experiment details for the reranking task mentioned in Section 1, where we analyze specific examples within the task.

## A.1 Pseudocode of RLMRec

This section introduces the pseudocode for our model-agnostic RLMRec framework implementations, namely RLMRec-Con and RLMRec-Gen. The focus is on the training procedure of these implementations. Prior to training, user and item profiles are preprocessed, and their semantic embeddings $\mathbf{s}$ are generated using text models. Algorithm 1 presents the training procedure for RLMRec-Con, while Algorithm 2 outlines the process for RLMRec-Gen.

The difference between RLMRec-Con and RLMRec-Gen is that RLMRec-Gen randomly masks a portion of users/items before the base model encodes the CF-side relational representations. The objective function $\mathcal{L}_{info}$ for mutual information maximization is then computed based on the representations of the masked users and items. In contrast, RLMRec-Con models the density ratio in a contrastive manner and calculates the $\mathcal{L}_{info}$ objective for all users and items in the batch data, including both positives and negatives.

## A.2 Details of Profile Generation

In this section, we offer a comprehensive explanation of the generation process for both user and item profiles. Real examples from the Amazon-book dataset are used to illustrate this process, as depicted in Figure 7 and Figure 8. We adopt a general interaction paradigm with large language models (LLMs), where the system prompt serves as an instruction to guide the profile generation task. While the Amazon-book dataset is specifically showcased, the overall generation process remains consistent for the Yelp and Steam datasets as well, with minor differences in the instructions provided to represent the data information.

*A.2.1* **Example of the Generated Item Profile.** Figure 7 showcases an example of item profile generation specifically for the Amazon-book dataset. The instruction prompt provided to the language models for all items remains consistent, directing them to summarize the types of books that would appeal to users, thus offering valuable information for recommendation purposes. The input information consists of the book's title and original description from the dataset. To maintain consistency and facilitate parsing, we enforce the requirement that the output of the language models adhere to the JSON format. Furthermore, it is essential for the language models to provide their reasoning behind the generated profile, ensuring high-quality summarization while preventing any potential hallucinations. The generated results demonstrate that the language model, in this case ChatGPT, accurately captures from the book description that the book is likely to attract readers interested in mental health and women's experiences.

*A.2.2* **Example of the Generated User Profile.** Figure 8 illustrates the process of generating user profiles using the Amazon-book dataset as an example. Our approach adopts an item-to-user generation paradigm, which allows us to leverage the previously generated profiles that describe the interaction preferences of items. To accomplish this, our prompt methodology incorporates not only users' feedback information on items but also the profiles of the items themselves. By utilizing both sources of information comprehensively, large language models are empowered to capture users' true preferences with enhanced accuracy. In the presented example, leveraging both the book descriptions and users' review text, the language models distill the user's preference for young adult fiction that seamlessly combines paranormal or supernatural elements.

## A.3 Analysis on the Reranking Task with LLMs

In this section, we explore real-world scenarios where LLMs are employed for reranking tasks on the Amazon-book dataset, as introduced in Section 1. Firstly, as depicted in Figure 9, we carefully design the instruction and input prompt. We utilize the item ID as

You will serve as an assistant to help me summarize **which types of users would enjoy a specific book**.
I will provide you with the title and a description of the book. Here are the instructions:
1. I will provide you with information in the form of a JSON string that describes the book:
{ "title": "the title of the book", (if there is no title, I will set this value to "None")
  "description": "a description of the book", (if there is no description, I will set this value to "None") }

**Requirements:**
1. Please provide your answer in **JSON format**, following this structure:
{ "summarization": "A summarization of what types of users would enjoy this book" (if you are unable to summarize it, please set this value to "None")
  "**reasoning**": "briefly explain your reasoning for the summarization" }
2. Please ensure that the "summarization" is no longer than 200 words.
3. Please ensure that the "reasoning" is no longer than 200 words.
4. Do not provide any other text outside the JSON string.

*Instruction*

{ "title": "The Bell Jar: A Novel (Perennial Classics)",
  "description": "Plath was an excellent poet but is known to many for this largely autobiographical novel. The Bell Jartells the story of a gifted young woman's mental breakdown beginning during a summer internship as a junior editor at a magazine in New York City in the early 1950s. The real Plath committed suicide in 1963 and left behind this scathingly sad, honest and perfectly-written book, which remains one of the best-told tales of a woman's descent into insanity.--This text refers to the Hard coveredition. "}

*Input Prompt*

{ "summarization": "The Bell Jar would appeal to those interested in reading about **mental health and women's experiences**. Specifically, readers who **enjoy raw and honest depictions of mental illness and its effects on a young woman's life** would appreciate this book.",
  "reasoning": "The Bell Jar delves into the mental breakdown of a young woman and her experiences navigating mental health and societal expectations as a woman in the 1950s. The book's autobiographical nature and raw, honest depiction of mental illness make it a compelling read for those interested in exploring these themes. Additionally, readers looking for works that examine the intersection of gender and mental illness would find The Bell Jar particularly thought-provoking." }

*Generated Item Profile*

**Figure 7: Case study on item profile generation in Amazon-book data.**

You will serve as an assistant to help me determine **which types of books a specific user is likely to enjoy**.
I will provide you with information about books that the user has purchased, as well as his or her reviews of those books.
Here are the instructions:
1. Each purchased book will be described in JSON format, with the following attributes:
{ "title": "the title of the book", (if there is no title, I will set this value to "None")
  "**description**": "a description of **what types of users will like this book**",
  "review": "the user's review on the book" (if there is no title, I will set this value to "None") }
2. The information I will give you:
**PURCHASED ITEMS**: a list of JSON strings describing the items that the user has purchased.

**Requirements:**
1. Please provide your decision in **JSON format**, following this structure:
{ "summarization": "A summarization of what types of books this user is likely to enjoy" (if you are unable to summarize it, please set this value to "None")
  "**reasoning**": "briefly explain your reasoning for the summarization" }
2. Please ensure that the "summarization" is no longer than 100 words.
3. The "reasoning" has no word limits.
4. Do not provided any other text outside the JSON string.

*Instruction*

PURCHASED ITEMS: [
{ "title": "Croak",
  "description": "**Young adult readers who enjoy paranormal and fantasy themes** would enjoy Croak.",
  "review": "**Loved the writing style**, was different than most of what I have read. The narrative was like a storyteller, you could hear someone telling the story to you. …"}

{ "title": "Deadly Cool (Hartley Featherstone)",
  "description": "Teenage girls who **enjoy a mix of humor, mystery, and high school drama** would enjoy Deadly Cool by Gemma Halliday. With plenty of red herrings and a quick pace, this book will keep …",
  "review": "**I really enjoyed reading this**, was laughing out loud in the middle of the night. ..."}

{ "title": "Stitch (Stitch Trilogy, Book 1)",
  "description": "**Fans of young adult paranormal romance novels** with a dash of mystery and suspense would enjoy Stitch (Stitch Trilogy, Book 1).",
  "review": "…. **Book started out really well, had me totally hooked from the start**. Love me a good ghost story, …. "}
… (*Omitted due to page limit*) ]

*Input Prompt*

{ "summarization": "This user **enjoys young adult fiction that** blends paranormal or supernatural elements with romance, mystery, humor, and coming-of-age themes. They also appreciate stories with complex world-building … .",
  "reasoning": "Based on the reviews and descriptions of the purchased items, **the user seems to be drawn to young adult fiction that features paranormal or supernatural elements, such as ghosts and magical powers.** They also enjoy a mix of genres, including romance, mystery, humor, and coming-of-age themes. … ." }

*Generated User Profile*

**Figure 8: Case study on user profile generation in Amazon-book data.**

You serve as an assistant that helps me to recommend 20 items (each item is a book) to a specific user from a candidate item set with the size of 40.
First, I will provide you with a list of items that he or she has already interacted with.
Then, I will provide you with a candidate item set which includes the index and title of each item.
Your goal is to create a list of recommended items from the candidate item set for the user, and the more recommended items are in the front of the list.
Here are the instructions:
1. Each item in the candidate set will be described in JSON format, with the following attributes:
{
    "index": "a unique integer to identify the item",
    "title": "the title of the item (book)"
}

2. The information I will give you:
INTERACTED ITEMS: a list of titles of items that the user has interacted before.
CANDIDATE ITEMS: a list of JSON strings that contains the candidate items.

Requirements:
1. When recommending, please explore his/her preferences through the items that the user has interacted with to determine the recommendation list.
2. Recommended items can only be selected from the CANDIDATE ITEMS.
3. Please provide your decision in JSON format, following this structure:
{
    "items": "a list of index of the recommend items" (e.g., [23, 45, 68, ...])
    "reasoning": "briefly explain your reasoning for the decision"
}
4. Make sure there are 20 recommended items in the list and do not use the title to represent items in the list.
5. Do not provided any other text outside the JSON string.

**Instruction**

INTERACTED ITEMS: ['The Sea of Tranquility: A Novel', 'Killing Sarai (In the Company of Killers)', 'Hopeless', 'The Raven Boys (Raven Cycle)', 'Truth (Consequences)', 'How to Kill a Rock Star', 'Unteachable', 'Shadow and Bone (Grisha Trilogy)', 'Under Different Stars (The Kricket Series) (Volume 1)', 'Running in Place (Mending Hearts) (Volume 2)']
CANDIDATE ITEMS: [ …
{"index": 329, "title": "Point of Retreat"}
{"index": 207, "title": "Vain: Book One of The Seven Deadly Series (Volume 1)"}
{"index": 269, "title": "Onyx (Lux)"}
{"index": 597, "title": "Captive in the Dark: The Dark Duet"}
{"index": 340, "title": "Sins & Needles (The Artists Trilogy)"}
{"index": 208, "title": "Ten Tiny Breaths"}
{"index": 940, "title": "Angelfall (Penryn & the End of Days, Book 1)"}
… (*Omitted due to page limit*) ]

**Input Prompt**

{  "items": [585, 597, 330, 1098, 293, 344, 327, 340, 752, 1141, 1071, 1022, 329, 319, 2911, 1057, 1253, 721, 974, 1127],
    "reasoning": "Based on the user's previous interactions, it seems that they enjoy books with dark and intense themes, so I have selected books like 'Raw', 'Captive in the Dark', and 'Seduced in the Dark' that fit this genre. I have also included some books with a similar writing style like 'Thoughtless' and 'Slammed'. Additionally, I have chosen books from popular series such as 'Lux' and 'Divergent', which the user may be interested in based on their previous interactions."}

**Rerank Results**

Figure 9: Case study on LLMs-based reranking. The candidate items are retrieved by LightGCN.

a key to index the items, along with their textual information such as the book title. This approach aims to enhance the accuracy of the generated outputs by avoiding direct generation of item titles that may not precisely match the original titles.

In the prompt, we include the historical purchased books of the user as a reference, along with a list of 40 candidate items retrieved by LightGCN. The goal is for the LLM to select the top 20 items for recommendation. However, upon examining the output results, it becomes apparent that the language models have recommended some non-existent items (highlighted in green) within the provided list. Despite utilizing item IDs for indexing, this occurrence is common in many reranking examples, and the presence of non-existent items can undoubtedly impact the overall reranking performance.

Additionally, the number of correctly recommended items from the language models is lower than the retrieved items (highlighted

in red). This discrepancy is primarily attributed to the limited textual information available for the language models to effectively exploit users' preferences. Moreover, the retrieved item list, learned by the state-of-the-art method LightGCN, benefits from collaborative information beyond just the textual content. This collaborative information contributes to the improved performance of the retrieval process compared to the language models' recommendations.

Incorporating other raw textual information from the datasets to improve performance may have some anticipated limitations: i) The limitation of input token numbers may constrain the size of candidate items, as many raw descriptions can be excessively lengthy. ii) Raw descriptions may be missing or contain substantial noise in certain datasets. The absence of descriptions or the presence of noisy information can hinder the language models' comprehension of users' preferences. iii) Including a larger amount of input data, such as additional raw textual information, can increase the computational cost, which impacts the system's scalability.

