# OpenReview forum: "Representation Learning with Large Language Models for Recommendation"
_ACM.org/TheWebConf/2024/Conference — TheWebConf24_

### Official Review · Reviewer_o5UN · 2023-11-16

**Novelty:** 4
**Technical Quality:** 5

**Review:**

Pros:
Quality: The paper signifies a meaningful advancement in recommender systems by adeptly leveraging Large Language Models (LLMs). The RLMRec framework, its primary contribution, adeptly integrates LLMs into extant recommender systems. It aims to surmount the constraints inherent in traditional ID-based models. The framework's key strength is its focus on capturing complex semantic aspects through the advanced summarization capabilities of LLMs.
Originality: The fusion of LLMs with recommender systems for enriched representation learning is a pioneering move. The model-agnostic character of the RLMRec framework, facilitating its integration with diverse recommender models, is a testament to its innovative approach.
Significance: This work's importance is rooted in its potential to transform recommender system functionalities. By augmenting their ability to process and leverage textual data more effectively, it sets the stage for more refined and personalized recommendations.
Cons:
Inference Time and Efficiency: The use of pre-trained LLMs for data augmentation and contrastive learning raises questions about coherence with identified limitations. The inference process for user/item profiling is time-intensive, potentially dominating the training efficiency analysis.
Baseline Comparisons: The selection of baselines for comparison is somewhat limited. Comparing primarily with ID-based methods falls short of fully demonstrating the superiority of incorporating LLMs. Including hybrid methods that combine ID and text embeddings would provide a more comprehensive evaluation.
Noise Robustness: The paper’s claims regarding noise robustness are not clearly substantiated in the ablation study. This lack of clarity raises questions about the model's effectiveness in dealing with noisy data.

**Questions:**

1.	The paper demonstrates the improvement of graph-based methods with the proposed paradigm. However, it lacks a comparison with simpler approaches, such as the concatenation of pre-trained BERT embeddings. Can you elaborate on why the contrastive or generative alignment is more beneficial for utilizing semantic information in recommendations compared to these simpler methods?
2.	The ablation study does not convincingly demonstrate the superior noise resistance of RLMRec-Con compared to the base Light GCN model, as both show similar performance decay rates with increased noise. How do you justify the claim of enhanced noise handling capability in RLMRec-Con?

**Reviewer Confidence:**

3: The reviewer is confident but not certain that the evaluation is correct

**Scope:**

4: The work is relevant to the Web and to the track, and is of broad interest to the community

---

### Official Review · Reviewer_tubz · 2023-11-20

**Novelty:** 6
**Technical Quality:** 6

**Review:**

The paper introduces a model-agnostic framework called RLMRec, designed to enhance traditional ID-based recommender systems using large language models (LLMs). This new paradigm integrates representation learning with LLMs, enabling the capture of intricate semantic aspects of user behaviors and preferences.

**Strength**

1. The paper establishes a theoretical foundation, demonstrating that incorporating textual signals through mutual information maximization can enhance the quality of representations.

2. The proposed RLMRec framework enhances existing recommender systems with LLM-empowered representation learning. This framework can incorporate auxiliary textual signals while providing model-agnostic enhancement.

3.  In the evaluation, RLMRec shows effectiveness with multiple state-of-the-art recommender models, and its efficiency and robustness to noisy data are analyzed.

**Weakness**

1. Some steps in the derivation of Theorem 1 and 2 are not clear. A more comprehensive version should be included in the appendix.

**Questions:**

See weakness

**Ethics Review Description:**

No issue

**Reviewer Confidence:**

4: The reviewer is certain that the evaluation is correct and very familiar with the relevant literature

**Scope:**

4: The work is relevant to the Web and to the track, and is of broad interest to the community

---

### Official Review · Reviewer_fjPz · 2023-11-24

**Novelty:** 3
**Technical Quality:** 3

**Review:**

The quality of this article is quite good, with clear expression, but it lacks originality and is of moderate importance.

Pros:
1. It proposes a framework unrelated to the graph recommendation model, which can enhance the recommendation effect of the graph recommendation model with only a slight increase in consumption.
2. It uses the powerful semantic extraction features of LLMs to mine high-level text to enhance recommendation embeddings.
3. The experiments are well-rounded and support the points put forward.

Cons:
1. While the approach is indeed innovative, it appears that the novelty primarily lies in substitifying the module for extracting high-level features of users and items in previous works with LLMs.
2. There is a lack of comparison with other  model-agnostic enhancement methods.

**Questions:**

1. Would it be possible to compare more enhancement architectures that are unrelated to the model, in order to demonstrate the superiority of LLMs?
2. Could you please explain why we use LLMs for feature extraction and provide experimental validation to illustrate the difference between LLMs and traditional LMs?

**Reviewer Confidence:**

4: The reviewer is certain that the evaluation is correct and very familiar with the relevant literature

**Scope:**

4: The work is relevant to the Web and to the track, and is of broad interest to the community

---

### Decision · Program_Chairs · 2024-01-22

**Decision:**

Accept

**Comment:**

This work tries to improve recommender systems by integrating large language models with representation learning, to capture the intricate semantic aspects of user behaviors and preferences. The proposed approach is novel, and the direction seems valid. However, the paper could be improved by providing more details. More importantly, the evaluation could be much improved by selecting more relevant baselines.